# Postoperative Imaging and Tumor Marker Surveillance in Resected Pancreatic Cancer

**DOI:** 10.3390/jcm8081115

**Published:** 2019-07-27

**Authors:** Hsu Wu, Jhe-Cyuan Guo, Shih-Hung Yang, Yu-Wen Tien, Sung-Hsin Kuo

**Affiliations:** 1Department of Oncology, National Taiwan University Hospital, Taipei, Taiwan; 2Department of Oncology, National Taiwan University Hospital, Yun-Lin Branch, Yunlin, Taiwan; 3Graduate Institute of Clinical Medicine, National Taiwan University College of Medicine, Taipei, Taiwan; 4Cancer Research Center, National Taiwan University College of Medicine, Taipei, Taiwan; 5National Taiwan University Cancer Center, National Taiwan University College of Medicine, Taipei, Taiwan; 6Department of Surgery, National Taiwan University Hospital and National Taiwan University College of Medicine, Taipei, Taiwan; 7Graduate Institute of Oncology, National Taiwan University College of Medicine, Taipei, Taiwan

**Keywords:** pancreatic cancer, follow-up, imaging, CA19-9, CEA

## Abstract

**Background**: Pancreatic cancer is a catastrophic disease with high recurrence and death rates, even in early stages. Early detection and early treatment improve survival in many cancer types but have not yet been clearly documented to do so in pancreatic cancer. In this study, we assessed the benefit on survival resulting from different patterns of surveillance in daily practice after curative surgery of early pancreatic cancer. **Methods**: Patients with pancreatic ductal adenocarcinoma who had received curative surgery between January 2000 and December 2013 at our institute were retrospectively reviewed. Patients were classified into one of four groups, based on surveillance strategy: the symptom group, the imaging group, the marker group (carbohydrate antigen 19-9 and/or carcinoembryonic antigen), and the intense group (both imaging and tumor marker assessment). Overall survival (OS), relapse-free survival (RFS), and post-recurrence overall survival (PROS) were evaluated. **Results**: One hundred and eighty-one patients with documented recurrence or metastasis were included in our analysis. The median OS for patients in the symptom group, imaging group, marker group, and intense group were 21.4 months, 13.9 months, 20.5 months, and 16.5 months, respectively (*p* = 0.670). Surveillance with imaging, tumor markers, or both was not an independent risk factor for OS in univariate and multivariate analyses. There was no significant difference in median RFS (symptom group, 11.7 months; imaging group, 6.3 months; marker group, 9.3 months; intense group, 6.9 months; *p* = 0.259) or median PROS (symptom group, 6.9 months; imaging group, 7.5 months; marker group, 5.0 months; intense group, 7.8 months; *p* = 0.953) between the four groups. Multivariate analyses identified poor Eastern Cooperative Oncology Group Performance Status (ECOG PS) (≥1), primary tumor site (tail), and tumor grade (poor differentiation) were poor prognostic factors for OS. **Conclusions**: Surveillance with regular imaging, tumor marker, or both was not an independent risk factor for OS of pancreatic cancer patients who undergo curative tumor resection.

## 1. Introduction

Patients with pancreatic cancer have a poor prognosis, even those initially diagnosed with resectable disease [1,2,3]. Compared with observation alone, adjuvant chemotherapy with either gemcitabine (the CONKO-001 study) or fluorouracil increases recurrence-free survival (RFS) and overall survival (OS) [4,5]. The JASPAC 01 study demonstrated a significant improvement in OS comparing S-1 to gemcitabine as the adjuvant setting for pancreatic cancer [6]. The ESPAC-4 study further demonstrated the improved median OS with a combined regimen of gemcitabine and capecitabine than gemcitabine alone as adjuvant therapy for pancreatic cancer [7]. However, the median RFS in these studies was only 11 to 15 months for pancreatic cancer patients receiving adjuvant gemcitabine, fluorouracil, or gemcitabine plus capecitabine [4,5,6,7]. Recently, a phase III trial (the PRODIGE 24/CCTG PA.6 trial) found that adjuvant mFOLFIRINOX (a modified regimen containing oxaliplatin, leucovorin, irinotecan, and fluorouracil) led to a better median RFS and OS than adjuvant gemcitabine in patients with resected pancreatic ductal adenocarcinoma [8]. However, there were still 60% of patients in the mFOLFIRINOX arm that relapsed within 3 years [8].

The grave prognosis of pancreatic cancer may be attributed to its biological behavior of early metastatic spreading, sometimes even before clinical evidence of a pancreatic tumor [9,10,11], and its interaction with a complex microenvironment [11,12]. Without adjuvant radiotherapy, at least 30% of patients experience local recurrence [4,7], and more than 50% of patients recur with distant metastasis [4,5]. The current consensus about postoperative surveillance strategies lacks large-scale validation. Contrary to the National Comprehensive Cancer Network (NCCN; United States) guidelines and the American Society of Clinical Oncology (ASCO) clinical practice guidelines [13,14], which recommend the use of a combination of history taking and physical examination (H & P), laboratory tests for serum carbohydrate antigen 19-9 (CA19-9) level, and computed tomography (CT) of the abdomen and pelvis every 3–6 months for 2 years and then annually as routine surveillance for pancreatic cancer patients after surgical resection, guidelines by the European Society for Medical Oncology (ESMO), United Kingdom National Institute for Health and Care Excellence, and National Dutch Pancreatic Cancer Group do not recommend the use of additional examinations for surveillance of patients who have undergone radical resection of pancreatic cancer because of the lack of evidence that detected recurrence could improve the OS [15,16,17]. 

The surveillance strategy varies enormously across randomized trials for adjuvant therapy of resected pancreatic cancer. In the CONKO-001 study, examination of CA19-9 every 4 weeks and abdominal sonography every 8 weeks were included in the follow-up protocol during adjuvant treatment [4]. In the JASPAC 01 study, abdominal CT and chest radiography every 3 months were performed for imaging surveillance [6]. Tumor assessment was dependent on local practice in the ESPAC-4 study [7]. Regardless of the strategy used in these randomized trials, the median RFS and OS with adjuvant gemcitabine were consistently around 11 to 13 months and 22 to 25 months, respectively [4,6,7]. In the PRODIGE 24/CCTG PA.6 study, tumor assessment was performed by CT every 3 months [8]. The RFS of the gemcitabine arm in the PRODIGE 24/CCTG PA.6 study was in line with previous studies, but the median OS was longer [8], which was probably due to the improvement of treatment for patients with metastatic disease.

The best strategy of postoperative surveillance is not well-defined and not readily inferred from the aforementioned published trials of adjuvant therapy. An even greater debate comes from the concepts of “regular” and “symptom-driven” assessment in real-world practice. A retrospective study using the Surveillance, Epidemiology, and End Results–Medicare linked data revealed no temporal pattern in CT use, reflecting symptom-driven examination rather than regular surveillance in the United States [18]. However, a retrospective study demonstrated shorter post-recurrence survival in patients with symptomatic recurrence than in those with asymptomatic recurrence, because asymptomatic patients received treatment more frequently [19]. 

Most clinical practice guidelines for resected pancreatic cancer patients recommend the use of a combination of tumor marker tests and CT in addition to H & P [20,21]. However, the aforementioned guidelines are mostly based on the expert opinion of physicians and category 2B evidence. Currently, little is known about the clinical impact on prognosis of the varying postoperative surveillance strategies for resected pancreatic cancer. The purpose of this study is to evaluate the patterns of surveillance in daily practice and their association with prognosis in pancreatic cancer patients who undergo curative tumor resection.

## 2. Materials and Methods

### 2.1. Patient Selection

Pancreatic cancer patients who underwent curative surgery between January 2000 and December 2013 were selected from the Cancer Registry, Medical Information Management Office of the National Taiwan University Hospital in Taipei, Taiwan. Other inclusion criteria were available medical records, pathologically confirmed invasive pancreatic cancer, and no neoadjuvant therapy. Neuroendocrine tumors and invasive tumors other than adenocarcinoma were excluded. Patients who did not experience recurrence were also excluded. Considering that patients who had recurrence within three months after surgery may have aggressive biological behavior, and all of them underwent the second-line treatment, including systemic chemotherapy or local treatment, we excluded patients who developed recurrence within three months after resection (all patients underwent the first follow-up within three months after resection, regardless of the surveillance strategy) in this study to avoid bias in assessing the association between different surveillances and clinical outcomes. The study protocol was approved by the Research Ethical Committee of National Taiwan University Hospital (201405031RINB).

In this retrospective study, patients with pancreatic cancer who underwent curative surgery from January 2000 to December 2013 (14 years) were analyzed. Before 2013, the follow-up strategy for pancreatic cancer patients after surgical resection varied with the era and with the physician’s clinical practice, as the evidence of surveillance using CA19-9 and CT for resected pancreatic cancer patients are classified under category 2B by the NCCN guidelines. H & P is the standard surveillance after surgical resection of pancreatic cancer based on the current NCCN and ASCO clinical practice guidelines [13,14], and the serum CA19-9 test and CT are potentially beneficial to detect early recurrence after resection [19,20,21]. We classified the resected pancreatic cancer patients into four groups: (1) symptom group, clinical evaluation alone with H & P; (2) imaging group, combined H & P and imaging (either CT or magnetic resonance imaging); (3) marker group, combined H & P and tumor marker tests, including tests for CA19-9 and/or carcinoembryonic antigen (CEA); (4) intense group, combined H & P, tumor marker tests, and imaging. 

Because of the reimbursement policy in Taiwan, tumor markers (CEA and CA19-9) and imaging could be arranged within three months, so we defined four months as the cut-off point for classification of patients based on the follow-up into regular and irregular. We defined “regular follow-up” as clinical evaluation, including imaging or tumor markers, assessed on average every 4 months or more frequently before recurrence or within 2 years after surgery if recurrence did not occur in the first 2 years. The date of death was obtained from the medical records in our institution or from the death registration of Taiwan. Recurrence was documented by imaging. The cut-off normal values of CA19-9 and CEA in our institution were 37 U/mL and 5 ng/mL, respectively.

### 2.2. Statistical Analysis

We defined OS as the duration from the date of surgery to the date of death from any cause or censoring if patients remained alive at the last follow-up. RFS was defined as the duration from the date of surgery to the date of documented recurrence. Patients were censored if recurrence had not occurred by the last follow-up. Post-recurrence overall survival (PROS) was defined as the duration from the date of recurrence to the date of death from any cause or censoring if patients remained alive at the last follow-up.

In this study, the primary endpoint was to evaluate the association between OS and four postoperative surveillance strategies. In addition, we assessed the association between RFS and PROS and four postoperative surveillance strategies. The survival of OS, RFS, and PROS was estimated using the Kaplan–Meier method and compared using the log-rank test. Considering that historical data of our patients was obtained from 2000 to 2013—and there may have been many practice-changing advances in more recent years—we further assessed the association between OS, RFS, and PROS and four postoperative surveillance strategies for patients diagnosed between January 2000 and July 2008, and August 2008 and December 2013, respectively.

To compare characteristics among the subgroups, the chi-square test was used for categorical variables and the Kruskal–Wallis test for continuous variables. Univariate analysis and multivariate analysis were performed to identify independent prognostic factors, including adjuvant chemotherapy (yes or not), performance status (0 vs. ≥1), primary site (non-tail vs. tail), surgical section margin (positive vs. negative), tumor (T) stage (T1/2 vs. T3), lymph node (N) stage (N1 vs. N0), grade (1/2 vs. grade 3), treatment after recurrence (yes or not), and postoperative surveillance strategies. A *p* value less than 0.05 was defined as significant for primary endpoint. 

## 3. Results

### 3.1. Clinicopathological Features and Adjuvant Treatment 

Our initial patient cohort included 319 patients with pancreatic ductal adenocarcinoma who underwent curative resection between 2000 and 2013. Ninety-four patients (29.5%) did not experience recurrence, and 44 patients experienced recurrence within 3 months after surgery. After excluding these patients, a total of 181 patients were included for further analysis, as shown in Figure 1. The characteristics of the four surveillance subgroups are listed in Table 1. The surgical margin was positive in 38 (21%) of patients. Regional lymph node metastasis was found in 90 (49.7%) patients. The median levels of CEA and CA 19-9 after operation were 1.43 ng/mL (range, 0.1 to 110 ng/mL) and 31.1 U/mL (range, 0.1 to 16,453.3 U/mL), respectively. Adjuvant chemotherapy was administered in 55 patients (30.4%), and the agents used were fluorouracil in 24 patients, gemcitabine in 22 patients, and sequential gemcitabine and fluorouracil in 9 patients. The median time from surgery to the start of adjuvant chemotherapy was 7 weeks (range, 3.4 to 28.6 weeks), as shown in Appendix A. Adjuvant radiotherapy was administered in 22 (12.2%) patients. None of these clinical characteristics before or after surgery differed significantly among the subgroups, except that adjuvant chemotherapy was administered most frequently in the intense subgroup and most rarely in the symptom subgroup (*p* < 0.001).

### 3.2. The Follow-Up and the Treatment Outcome of Four Surveillance Groups

The follow-up intervals of each group are listed in Appendix A. The detailed pattern of surveillance in the symptom group is provided in Figure 2. The median follow-up for the whole population was 16.0 months. The median OS for the entire patient cohort was 17.8 months (95% confidence interval (CI): 15.4 to 20.3). The median OS was not significantly different (*p* = 0.670) among the four subgroups, with OS of 21.4 months (95% CI: 17.9 to 25.0), 13.9 months (95% CI: 6.5 to 21.4), 20.5 months (95% CI: 13.5 to 27.5), and 16.5 months (95% CI: 14.9 to 18.1) in the symptom group, imaging group, marker group, and intense group, respectively, as shown in Figure 3A and Table 2. The OS was not different in four subgroups of patients diagnosed between January 2000 and July 2008, as shown in Figure 4A. Similarly, for patients diagnosed between August 2008 and December 2013, the OS was not different in four subgroups, as shown in Figure 4B.

Overall, the median RFS was 8.1 months (95% CI: 7.2 to 9.1). The median RFS of the four subgroups are listed in Table 2, and the Kaplan–Meier curves are in Figure 3B. The intense group had significantly shorter median RFS than the symptom group (6.9 months versus 11.7 months, log-rank *p* = 0.001). There was no significant difference among the four subgroups in terms of PROS (*p* = 0.953), as shown in Table 2. Patients with postoperative elevation of CA19-9 or CEA had a poorer OS. Seven patients with postoperative elevation of both CA19-9 and CEA had the lowest OS, of only 7.8 months. Elevation of CA19-9 and CEA was noted before documented recurrence in 96 (53.0%) and 27 (14.9%) patients, respectively. The median times from operation to the elevation of CA19-9 and CEA were 5.1 and 5.7 months, respectively. The median times from elevation of CA19-9 and CEA to documented recurrence were 2.1 (range, 0 to 31.38) months and 1.4 (range, 0.03 to 16.52) months, respectively. 

### 3.3. Salvage Treatment for Recurrent Patients

The first-line salvage chemotherapy regimens included gemcitabine, gemcitabine with fluorouracil, oral fluoropyrimidine (S-1 or capecitabine), oxaliplatin with fluorouracil, gemcitabine with oxaliplatin and fluorouracil, and fluorouracil alone, and did not differ significantly among the four subgroups. The majority of the patients (number = 131, 72.4%) had distant metastasis at the first recurrence. The pattern of recurrence did not differ significantly among the four subgroups (*p* = 0.175), as shown in Table 1. After recurrence, patients who received treatment had a better median PROS (8.2 months, 95% CI: 6.1 to 10.2) than those who did not receive treatment (4.3 months, 95% CI: 2.1 to 6.6, *p* = 0.024). After recurrence, patients who received treatment earlier (≤4 weeks after recurrence) had a no better median PROS than those who received treatment later (>4 weeks after recurrence; 6.9 months (95% CI: 4.0 to 9.7) versus 10.4 months (95% CI: 8.4 to 12.5), *p* = 0.373).

### 3.4. Prognostic Factors

In the univariate analysis, positive lymph node metastasis (*p* = 0.043) and poor differentiation (*p* = 0.051) were related to shorter OS, whereas an Eastern Cooperative Oncology Group Performance Status (ECOG PS) of 1 had a trend of correlation with shorter OS (*p* = 0.064), as shown in Table 3. In the multivariate analysis, independent risk factors that influenced OS were ECOG PS (0 versus ≥1, HR = 0.516, *p* = 0.020), primary tumor site (head/body versus tail, HR = 0.599, *p* = 0.049), and tumor grade (good/moderate differentiation versus poor differentiation, HR = 0.553, *p* = 0.032). Follow-up strategy was not an independent risk factor for OS in either univariate or multivariate analyses, as shown in Table 3.

## 4. Discussion

In this study, we retrospectively analyzed an early pancreatic cancer cohort over 14 years at our institute. We found that intense follow-up with imaging and tumor markers after resection of early pancreatic cancer did not offer a survival benefit. Elevation of CA19-9 and CEA occurred shortly before documented recurrence by imaging. Moreover, beginning first-line treatment early (≤4 weeks after recurrence) for recurrence of pancreatic cancer did not result in a better PROS in our cohort. To the best of our knowledge, this is the first study to examine the relationship between different surveillance strategies and OS and PROS. 

Many studies recommend regular and structured follow-up after curative resection of pancreatic cancer, because asymptomatic recurrence is detected in up to 75% of patients [19,20,21,22,23,24]. Asymptomatic patients are more likely to receive further treatment, and prognosis of these patients will potentially be better [19]. One prospective study [22] on the cost-effectiveness of follow-up demonstrated the survival benefit of surveillance with clinical evaluation and CA19-9 testing every 6 months compared to no scheduled surveillance in patients who received neoadjuvant therapy and pancreaticoduodenectomy for pancreatic ductal adenocarcinoma. However, more intensive surveillance with increased frequency or combined with radiographic study increased cost but offered no survival benefit [22]. 

In the current study, we found that surveillance with combined H & P, tumor marker tests, and imaging was not associated with RFS, PROS, and OS of pancreatic cancer patients who underwent curative surgery. The possible reasons are: (1) the detection methods, including serum marker tests and imaging, were not sufficiently sensitive to detect the preneoplastic lesions of recurrent pancreatic cancer early, and as reported by recent studies, circulating pancreatic cells and tumor DNA, which can be detected in the bloodstream before tumor formation, could have been helpful [25,26,27]; (2) the treatment methods used in the period from 2000 to 2013, including systemic chemotherapy and local chemoradiotherapy, were not sufficiently effective to eradicate the recurrent pancreatic cancer, compared to the current systemic regimens, such as FOLFIRINOX, nab-paclitaxel, and nanoliposomal irinotecan, even when the recurrent disease was detected early [28,29,30,31,32,33]; (3) the underlying tumor biology and epigenetic and genetic factors of the pancreatic cancer lessened the treatment efficacies of systemic chemotherapy, molecular target agents, and radiotherapy, and thus, contributed to tumor progression, even when the disease was detected early [34,35,36]; (4) the clinicopathological features and patient factors affected the prognosis of the resected pancreatic cancer [37,38,39,40], as in our study, showing the primary site as the pancreatic tail, poor tumor differentiation, and poor patient performance. Birnbaum et al. revealed that gene expression signatures are different between the head (genes more associated with lymphocyte activation and pancreatic exocrine functions) and the body/tail (genes more associated with keratinocyte differentiation) of the pancreas [40].

Pancreatic cancer is well known to be a systemic disease early in the disease course [9,10,11]. Therefore, detecting disease as early as possible for resected pancreatic cancer becomes important. Motoi et al. reported that persistent elevation of serum CA19-9 is a risk factor for hepatic recurrence and is associated with poor prognosis [41]. Our study also showed that patients with postoperative elevation of CA19-9 had a poorer OS. Rieser et al. demonstrated that patients with initial elevation of CA19-9 and normalization after resection, followed by persistent elevation, or those with persistently elevated CA19-9 from diagnosis through surveillance were significantly associated with poor RFS and OS compared to those without elevated CA19-9 at initial diagnosis and persistent or periodic normalization after resection [42]. Rieser et al. also showed that elevation of CA19-9 preceded the recurrence examined on CT by over 6 months. In addition to CA19-9, serum CEA can be detected early compared to detection of lesions on CT in pancreatic cancer patients after surgical resection [43]. Reitz et al. reported that a combination of CEA and CA19-9 compared to CEA or CA19-9 alone improved the prognostic prediction in OS of stage I-III pancreatic cancer patients [44]. Xu et al. showed that among pancreatic cancer patients with post-resection normalization of CA19-9, elevated postoperative CEA was an independent risk factor for poor OS [45]. Our study also showed that seven patients with postoperative elevation of both CA19-9 and CEA had the lowest OS, which was of only 7.8 months. These findings suggest that if patients have elevated CA19-9 or CEA at the initial diagnosis, using a combination of CA19-9 and CEA for surveillance may help the early use of salvage therapy if the tumor markers show persistent elevation or normalization followed by elevation, and thereby improve patient survival. Circulating tumor DNA (ctDNA) has been reported to have prognostic value [25]. Pancreatic cancer patients with ctDNA after curative surgery had shorter disease-free survival and OS [25]. The surveillance of ctDNA after resection may play a role in detecting early occult recurrence or metastases of pancreatic cancer and treating them sooner, although this requires further study to determine whether this provides a subsequent gain in survival benefit.

Panels with a combination of multiple serum markers might improve recurrence detection. Many studies focused on combination panels for pancreatic cancer screening, such as combining carcinoembryonic antigen-related cell adhesion molecules (CEACAMs) [46,47,48], osteopontin [49], or matrix metallopeptidase 7 with CA19-9 [50], or combining CEA, matrix metalloproteinases 1 (TIMP-1), and CA19-9 [51] or haptoglobin, serum amyloid A, and CA19-9 [52]. All these combination panels demonstrated improved diagnostic accuracy, but whether they could be applied to postoperative surveillance needs further investigation [53]. In addition to tumor markers, two retrospective studies showed that under CT-based surveillance for resected pancreatic cancers, patients with asymptomatic recurrence underwent more post-recurrence treatments and showed better OS compared to those with symptomatic recurrence [19,54]. These studies indicated that imaging can detect asymptomatic recurrence of resected pancreatic cancers early, and these patients have relatively good performance status and tumor biology, and thus, benefit from subsequent treatments. 

In this study, the proportions of local recurrence, which is potentially more curable, were the same in the four strategy groups. However, PROS did not significantly differ between the early and late treatments, possibly because most patients underwent systemic treatment, such as with fluorouracil, oral fluoropyrimidine (S-1 or capecitabine), gemcitabine, or oxaliplatin, as salvage regimens. In metastatic pancreatic cancer patients, current studies have demonstrated that FOLFIRINOX, nab-paclitaxel plus gemcitabine, and nanoliposomal irinotecan increased the tumor responses and OS compared to the aforementioned conventional chemotherapy regimens [28,29,30,31,32,33]. In addition, several studies showed that aggressive treatment of an isolated local recurrence with re-resection, chemoradiotherapy, or stereotactic body radiation therapy provides survival benefit to these patients [55,56,57,58]. We believe that the use of a combination of H & P, tests for both tumor makers (CEA and CA19-9), and CT as surveillance can detect the local recurrence or metastases in resected pancreatic cancer patients early, and thus, allow aggressive local therapy and current systemic chemotherapy regimens, which will not only increase PROS (more than our PROS of 7.8 months in the intensive group) but also prolong OS. 

The limitations of our study are that patients with pancreatic ductal adenocarcinoma were diagnosed through a long period of time in this retrospective study. During such a long period of time, there have been many practice-changing advances, such as the improvement of chemotherapy regimens, during this period. To avoid potential bias, we divided our patients into two generations by July 2008; at that time CONKO-001 was published and gemcitabine became the standard treatment as adjuvant setting for resected pancreatic cancer. We found that the OS was not different in four subgroups of patients diagnosed between January 2000 and July 2008 and of those diagnosed between August 2008 and December 2013. Because our patients were grouped into one of four categories according to follow-up strategy, there were potential confounding factors to influence survival. Moreover, the choices of follow-up strategy were potentially highly biased by physicians’ habits, which we could not properly analyze. Nevertheless, we analyzed all factors that potentially influenced survival at baseline and found that only adjuvant chemotherapy was unbalanced between four strategies (more frequent use in the intense group). Through multivariate analysis, we found that ECOG performance status, primary tumor site, and tumor grade influenced OS, but adjuvant chemotherapy and follow-up strategy did not affect OS.

## 5. Conclusions

Our study indicated that early detection and early treatment of recurrent pancreatic cancer do not improve survival. Surveillance with assessment of signs and symptoms of recurrence every 3 to 6 months for 2 years after resection of pancreatic cancer remains the standard surveillance strategy [13,14,59,60]. Further studies to document the proper follow-up strategy in this era with new systemic and local treatments and to create more accurate methods to detect pancreatic cancer are warranted.

## Figures and Tables

**Figure 1 jcm-08-01115-f001:**
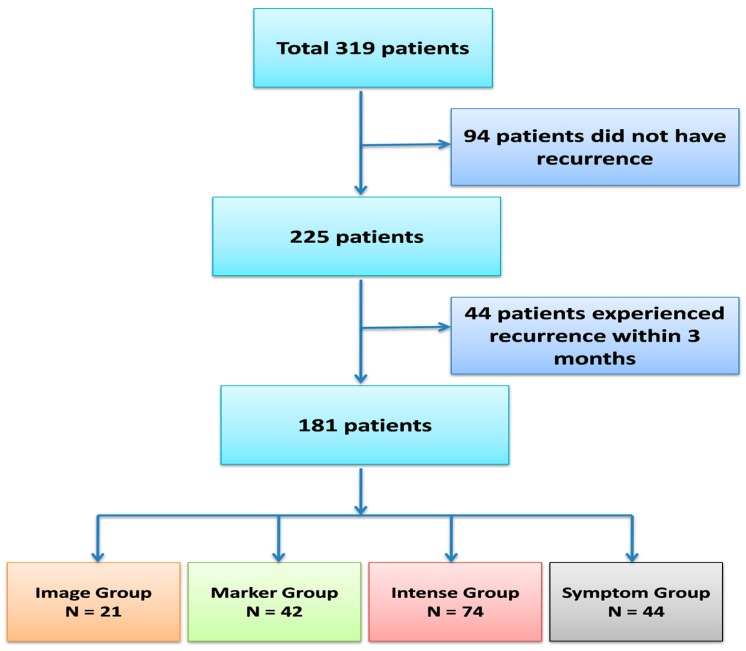
Consort diagram of resected pancreatic cancer patients.

**Figure 2 jcm-08-01115-f002:**
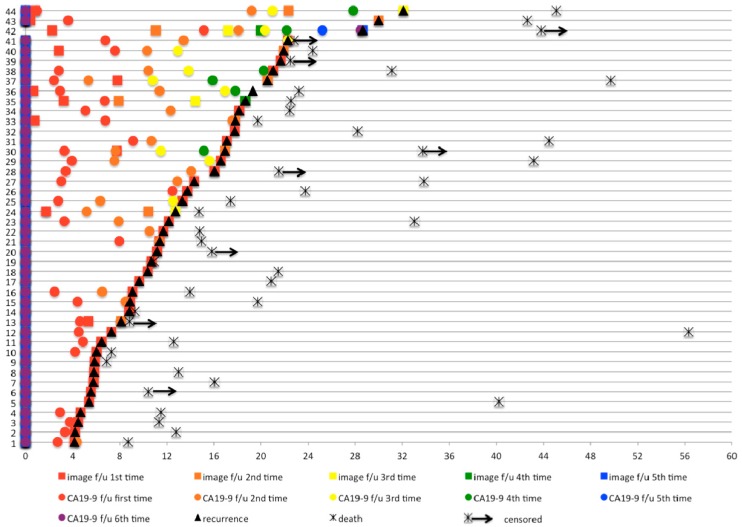
Postoperative follow-up (f/u) in the symptom group.

**Figure 3 jcm-08-01115-f003:**
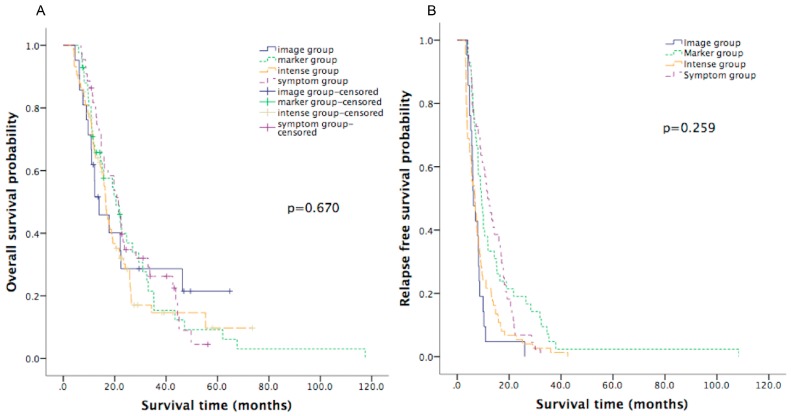
Kaplan–Meier curve of overall survival (**A**) and relapse-free survival (**B**) in the four strategy groups.

**Figure 4 jcm-08-01115-f004:**
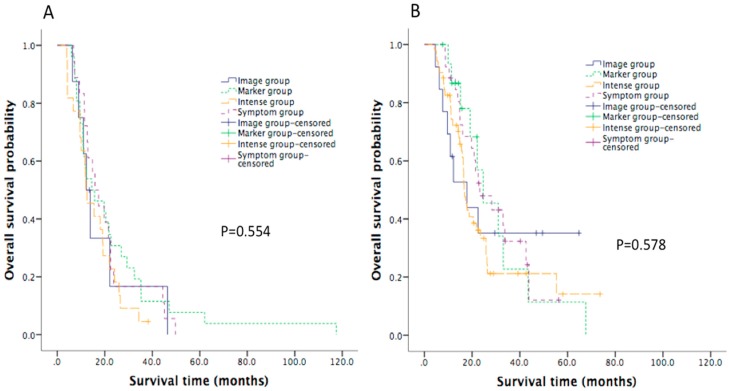
Kaplan–Meier curve of overall survival in patients diagnosed before July 2008 (**A**) and diagnosed after July 2008 (**B**) in the four strategy groups.

**Table 1 jcm-08-01115-t001:** Characteristics of all patients.

8		Symptom Group (*n* = 44)		Imaging Group (*n* = 21)		Marker Group (*n* = 42)		Intense Group (*n* = 74)		*p*-Value
		*N*	%	*N*	%	*N*	%	*N*	%	
**Age**	Median	72.5		70.3		65.1		66.0		0.069 ^¶^
**Gender**	Male	24	54.5%	14	66.7%	29	69.0%	44	59.5%	0.518
	Female	20	45.5%	7	33.3%	13	31.0%	30	40.5%	
**ECOG PS**	0	3	6.8%	3	14.3%	7	16.7%	13	17.6%	0.511
	1	34	77.3%	14	66.7%	32	76.2%	55	74.3%	
	2	7	15.9%	4	19.0%	3	7.1%	5	6.8%	
	3	0	0.0%	0	0.0%	0	0.0%	1	1.4%	
**Primary site**	Head	37	84.1%	15	71.4%	33	78.6%	55	74.3%	0.900
	Body	2	4.5%	2	9.5%	3	7.1%	5	6.8%	
	Tail	5	11.4%	4	19.5%	6	14.3%	14	18.9%	
**Section margin**	Negative	33	75.0%	15	71.4%	36	85.7%	59	79.7%	0.510
	Positive	11	25.0%	6	28.6%	6	14.3%	15	20.3%	
**pT**	T1	4	9.1%	2	9.5%	1	2.4%	0	0.0%	0.198
	T2	6	13.6%	3	14.3%	5	11.9%	9	12.2%	
	T3	34	77.3%	16	76.2%	36	85.7%	65	87.8%	
	T4	0	0.0%	0	0.0%	0	0.0%	0	0.0%	
**pN**	N0	24	54.5%	10	47.6%	23	54.8%	34	45.9%	0.736
	N1	20	45.5%	11	52.4%	19	45.2%	40	54.1%	
**Differentiation**	Grade 1	9	20.5%	4	19.0%	10	23.8%	10	13.5%	0.764
	Grade 2	30	68.2%	16	76.2%	29	69.0%	55	74.3%	
	Grade 3	5	11.4%	1	4.8%	3	7.1%	9	12.2%	
**Adjuvant C/T**	No	40	90.9%	18	85.7%	27	64.3%	41	55.4%	<0.001 *
	Yes	4	9.1%	3	14.3%	15	35.7%	33	44.6%	
**Pattern of first R**	Distant	27	61.4%	14	66.7%	34	81.0%	56	75.7%	0.175
	Local	17	38.6%	7	33.3%	8	19.0%	18	24.3%	
**Treatment after R**	No	24	54.5%	8	38.1%	17	40.5%	24	32.4%	0.130
	Yes	20	45.5%	13	61.9%	25	59.5%	50	67.6%	
**CA199 post OP (U/mL)**	Median	19.5		37.85		24.8		38.4		0.149 ^¶^
	Elevated	7	41.2%	5	55.6%	17	44.7%	37	50.7%	0.827
	Normal	10	58.8%	4	44.4%	21	55.3%	36	49.3%	
**CEA post OP (ng/mL)**	Median	2.45		0.91		1.26		1.52		0.042 ^¶^ *
	Elevated	2	11.1%	0	0.0%	1	2.9%	6	9.2%	0.539
	Normal	16	88.9%	6	100.0%	33	97.1%	59	90.8%	
**CA199 when R (U/mL)**	Median	209.71		1011.95		501		374.4		0.521^¶^
	Elevated	27	81.8%	9	75.0%	30	81.1%	55	78.6%	0.950
	Normal	6	18.2%	3	25.0%	7	18.9%	15	21.4%	
**CEA when R (ng/mL)**	Median	3.0		2.89		2.45		2.3		0.435 ^¶^
	Elevated	12	40.0%	2	22.2%	10	33.3%	15	23.4%	0.366
	Normal	18	60.0%	7	77.8%	20	66.7%	49	76.6%	

Abbreviations: T, tumor stage; N, lymph node stage; ECOG PS, Eastern Cooperative Oncology Group Performance Status (Grade 0, fully active, able to carry on all pre-disease performance without restriction; Grade 1, restricted in physically strenuous activity but ambulatory and able to carry out work of a light or sedentary nature, e.g., light house work, office work; Grade 2, ambulatory and capable of all selfcare but unable to carry out any work activities, up and about more than 50% of waking hours; Grade 3, capable of only limited selfcare, confined to bed or chair more than 50% of waking hours); C/T, chemotherapy; R, recurrence; OP, operation. ^¶^ Kruskal-Wallis test; * *p* < 0.05.

**Table 2 jcm-08-01115-t002:** Overall survival, relapse free survival, and post recurrence overall survival in four strategy groups.

	Symptom Group	Imaging Group	Marker Group	Intense Group	*p*-Value
**Median OS** **(months)**	21.4 (95% CI: 17.9 to 25.0)	13.9 (95% CI: 6.5 to 21.4)	20.5 (95% CI: 13.5 to 27.5)	16.5 (95% CI: 14.9 to 18.1)	0.670
**Median RFS** **(months)**	11.7 (95% CI: 8.9 to 14.5)	6.3 (95% CI: 4.6 to 7.9)	9.3 (95% CI: 7.5 to 11.1)	6.9 (95% CI: 5.2 to 8.6)	0.259
**Median PROS** **(months)**	6.9 (95% CI: 2.9 to 10.9)	7.5 (95% CI: 1.2 to 13.9)	5.0 (95% CI: 3.0 to 7.0)	7.8 (95% CI: 5.6 to 10.0)	0.953

Abbreviations: OS, overall survival; RFS, relapse-free survival; PROS, post-recurrence overall survival; CI, confidence interval.

**Table 3 jcm-08-01115-t003:** Univariate and multivariate analysis of overall survival.

	Univariate Analysis			Multivariate Analysis		
	Hazard Ratio (HR)	95.0% CI for HR	*P*-Value	Hazard Ratio (HR)	95.0% CI for HR	*P*-Value
**Adjuvant chemotherapy** **Yes vs. No**	1.003	0.696–1.445	0.988	0.925	0.596–1.436	0.728
**ECOG PS** **0 vs. ≥1**	0.626	0.381–1.028	0.064	0.516	0.295–0.903	0.020 *
**Primary site** **Non-tail vs. tail**	0.930	0.599–1.445	0.747	0.599	0.360–0.997	0.049 *
**Section margin** **Positive vs. Negative**	1.230	0.817–1.852	0.322	1.128	0.736–1.729	0.581
**T stage** **T1/T2 vs. T3**	0.732	0.471–1.139	0.167	0.623	0.387–1.002	0.051
**N stage** **N1 vs. N0**	1.406	1.010–1.956	0.043 *	1.368	0.961–1.947	0.082
**Differentiation** **Grade 1/2 vs. Grade 3**	0.595	0.353–1.002	0.051	0.553	0.322–0.950	0.032 *
**Treatment after recurrence** **Yes vs. No**	0.958	0.682–1.347	0.806	0.996	0.675–1.469	0.983
**Follow up strategy**	0.985	0.831–1.167	0.862	0.996	0.837–1.184	0.960

Abbreviations: T, tumor stage; N, lymph node stage; ECOG PS, Eastern Cooperative Oncology Group performance status; * *p* < 0.05.

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
