# Peer review of "Postoperative Imaging and Tumor Marker Surveillance in Resected Pancreatic Cancer"

_jcm, 2019, doi:10.3390/jcm8081115_

Reviewer 1 Report

In their retrospective study the authors have convincingly demonstrated that the follow-up work-up is not a prognostic factor. The study is well written and combines several standard procedures for recurrence detection in pancreatic cancer. The results are presented in a clear fashion and the conclusions drawn are adequate.

 Author Response

Authors’ response:

We greatly appreciate you for your great efforts in reviewing our manuscript.

We have carefully checked the grammar and spelling in the whole manuscript.

Reviewer 2 Report

This study is well designed, analyzed, and presented.

Major problems: 

The standard practice for surveillance of patients with surgically resected pancreatic adenocarcinoma is a combination of history and physical examination, blood tests including serum CA 19-9, and imaging studies (typically CT scans), at least in the United States, according to NCCN Guideline.

Overall survival of this patient population is unlikely related to the method of surveillance, but more likely related to cancer biology and genetics, patient factors, effectiveness of treatment.

Author Response

This study is well designed, analyzed, and presented.

Major problems: 

(1)  The standard practice for surveillance of patients with surgically resected pancreatic adenocarcinoma is a combination of history and physical examination, blood tests including serum CA 19-9, and imaging studies (typically CT scans), at least in the United States, according to NCCN Guideline.

(2)  Overall survival of this patient population is unlikely related to the method of surveillance, but more likely related to cancer biology and genetics, patient factors, effectiveness of treatment.

Authors’ response:

Authors’ response to Q1:

We appreciate the reviewer’s comment. In accordance with your insightful suggestion, we have added new information in the “Introduction” section and in the “Materials and Methods” section as follows:

In the “Introduction” section (Page 2, Lines 64-73; Lines 93-96):

Contrary to the National Comprehensive Cancer Network (NCCN; United States) guidelines and the American Society of Clinical Oncology (ASCO) clinical practice guidelines [13,14], which recommend the use of a combination of history taking and physical examination (H&P), laboratory tests for serum carbohydrate antigen 19-9 (CA19-9) level, and computed tomography (CT) of the abdomen and pelvis every 3–6 months for 2 years and then annually as routine surveillance for pancreatic cancer patients after surgical resection, guidelines by the European Society for Medical Oncology (ESMO), United Kingdom National Institute for Health and Care Excellence, and National Dutch Pancreatic Cancer Group do not recommend the use of additional examinations for surveillance of patients who have undergone radical resection of pancreatic cancer because of the lack of evidence that detected recurrence could improve the OS [15-17].

Most clinical practice guidelines for resected pancreatic cancer patients recommend the use of a combination of tumor marker tests and CT in addition to H&P [20,21]. However, the aforementioned guidelines are mostly based on the expert opinion of physicians and category 2B evidence. Currently, little is known about the clinical impact on prognosis of the varying postoperative surveillance strategies for resected pancreatic cancer. The purpose of this study is to evaluate the patterns of surveillance in daily practice and their association with prognosis in pancreatic cancer patients who undergo curative tumor resection.

In the “Materials and Methods” section (Page 3, Lines 115-126):

In this retrospective study, patients with pancreatic cancer who underwent curative surgery from January 2001 to December 2013 (13 years) were analyzed. Before 2013, the follow-up strategy for pancreatic cancer patients after surgical resection varied with the era and with the physician’s clinical practice, as the evidence of surveillance using CA19-9 and CT for resected pancreatic cancer patients are classified under category 2B by the NCCN guidelines. H&P is the standard surveillance after surgical resection of pancreatic cancer based on the current NCCN and ASCO clinical practice guidelines [13,14], and serum CA19-9 test and CT are potentially beneficial to detect early recurrence after resection [19-21]. We classified the resected pancreatic cancer patients into four groups: (1) symptom group, clinical evaluation alone with H&P; (2) imaging group, combined H&P and imaging (either CT or magnetic resonance imaging); (3) marker group, combined H&P and tumor marker tests, including tests for CA19-9 and/or carcinoembryonic antigen (CEA); (4) intense group, combined H&P, tumor marker tests, and imaging.

 Authors’ response to Q2:

We agree that the clinicopathological features, biology and genetics of pancreatic adenocarcinoma, and effectiveness of the systemic and local treatments may affect OS, relapse-free survival (RFS), and post-recurrence-free survival (PRFS) of patients with resected pancreatic adenocarcinoma, regardless of the surveillance strategy. In accordance with your suggestion, we have added new information in the “Discussion” section (Page 9, Lines 282-294; Page 10, Lines 295-300, Lines 339-346; Page 11, Lines 347-352) as follows:

“In the current study, we found that surveillance with combined H&P, tumor marker tests, and imaging was not associated with RFS, PROS and OS of pancreatic cancer patients who underwent curative surgery. The possible reasons are: (1) the detection methods, including serum marker tests and imaging, were not sufficiently sensitive to detect the preneoplastic lesions of recurrent pancreatic cancer early, and as reported by recent studies, circulating pancreatic cells and tumor DNA, which can be detected in the bloodstream before tumor formation, could have been helpful [25-27]; (2) the treatment methods used in the period from 2001 to 2013, including systemic chemotherapy and local chemoradiotherapy, were not sufficiently effective to eradicate the recurrent pancreatic cancer, compared to the current systemic regimens, such as FOLFIRINOX, nab-paclitaxel, and nanoliposomal irinotecan, even when the recurrent disease was detected early [28-33]; (3) the underlying tumor biology and epigenetic and genetic factors of the pancreatic cancer lessened the treatment efficacies of systemic chemotherapy, molecular target agents, and radiotherapy, and thus, contributed to tumor progression, even when the disease was detected early [34-36]; (4) the clinicopathological features and patient factors affected the prognosis of the resected pancreatic cancer [37-40], as in our study, showing the primary site as the pancreatic tail, poor tumor differentiation, and poor patient performance. Birnbaum et al. revealed that gene expression signatures are different between the head (genes more associated with lymphocyte activation and pancreatic exocrine functions) and the body/tail (genes more associated with keratinocyte differentiation) of the pancreas [40].”

In this study, the proportions of local recurrence, which is potentially more curable, were the same in the four strategy groups. However, PROS did not significantly differ between the early and late treatments, possibly because most patients underwent systemic treatment, such as with fluorouracil, oral fluoropyrimidine (S-1 or capecitabine), gemcitabine, or oxaliplatin, as salvage regimens. In metastatic pancreatic cancer patients, current studies have demonstrated that FOLFIRINOX, nab-paclitaxel plus gemcitabine, and nanoliposomal irinotecan increased the tumor responses and OS compared to the aforementioned conventional chemotherapy regimens [28-33]. In addition, several studies showed that aggressive treatment of an isolated local recurrence with re-resection, chemoradiotherapy, or stereotactic body radiation therapy provides survival benefit to these patients [55-58]. We believe that the use of a combination of H&P, tests for both tumor makers (CEA and CA19-9), and CT as surveillance can detect the local recurrence or metastases in resected pancreatic cancer patients early, and thus, allow aggressive local therapy and current systemic chemotherapy regimens, which will not only increase PROS (more than our PROS of 7.8 months in the intensive group) but also prolong OS.

Reviewer 3 Report

In the present study, Wu et al have investigated the survival benefits of different types of monitoring (eg tumor marker, imaging, etc) in pancreatic cancer patients who underwent surgery at an early stage. Overall, the authors found no improvement in survival with these surveillance methods. The overall study is retrospective in nature and the manuscript is presented well. Please find below my comments that may help the authors to further improve:

1) The authors might want to better explain the rationale behind classifying the patients in the four groups? What was the basis of selection of these groups? Also, why those patients with recurrence within 3 months were excluded from the study, needs to be highlighted.

2) While the authors do touch on the current clinical practice of surveillance of these patients, it would be great if they could explain in more details the same. Also, since they don't find any survival benefit with tumor marker or imaging monitoring, they might want to explain in depth, their take on what strategies could help us in better monitoring of these patients. 

Author Response

In the present study, Wu et al have investigated the survival benefits of different types of monitoring (eg tumor marker, imaging, etc) in pancreatic cancer patients who underwent surgery at an early stage. Overall, the authors found no improvement in survival with these surveillance methods. The overall study is retrospective in nature and the manuscript is presented well.

Please find below my comments that may help the authors to further improve:

 Q1: The authors might want to better explain the rationale behind classifying the patients in the four groups? What was the basis of selection of these groups? Also, why those patients with recurrence within 3 months were excluded from the study, needs to be highlighted.

Q2: While the authors do touch on the current clinical practice of surveillance of these patients, it would be great if they could explain in more details the same. Also, since they don't find any survival benefit with tumor marker or imaging monitoring, they might want to explain in depth, their take on what strategies could help us in better monitoring of these patients.  

Authors’ response:

Authors’ response to Q1:

We appreciate the reviewer’s comment. In accordance with your insightful suggestion, we have added new information in the “Introduction” section (and in the “Material and Methods” section as follows:

In the “Introduction” section (Page 2, Lines 64-73; Lines 93-96)

Contrary to the National Comprehensive Cancer Network (NCCN; United States) guidelines and the American Society of Clinical Oncology (ASCO) clinical practice guidelines [13,14], which recommend the use of a combination of history taking and physical examination (H&P), laboratory tests for serum carbohydrate antigen 19-9 (CA19-9) level, and computed tomography (CT) of the abdomen and pelvis every 3–6 months for 2 years and then annually as routine surveillance for pancreatic cancer patients after surgical resection, guidelines by the European Society for Medical Oncology (ESMO), United Kingdom National Institute for Health and Care Excellence, and National Dutch Pancreatic Cancer Group do not recommend the use of additional examinations for surveillance of patients who have undergone radical resection of pancreatic cancer because of the lack of evidence that detected recurrence could improve the OS [15-17].

Most clinical practice guidelines for resected pancreatic cancer patients recommend the use of a combination of tumor marker tests and CT in addition to H&P [20,21]. However, the aforementioned guidelines are mostly based on the expert opinion of physicians and category 2B evidence. Currently, little is known about the clinical impact on prognosis of the varying postoperative surveillance strategies for resected pancreatic cancer. The purpose of this study is to evaluate the patterns of surveillance in daily practice and their association with prognosis in pancreatic cancer patients who undergo curative tumor resection.

In the “Material and Methods” section (Page 3, Lines 115-132)

In this retrospective study, patients with pancreatic cancer who underwent curative surgery from January 2001 to December 2013 (13 years) were analyzed. Before 2013, the follow-up strategy for pancreatic cancer patients after surgical resection varied with the era and with the physician’s clinical practice, as the evidence of surveillance using CA19-9 and CT for resected pancreatic cancer patients are classified under category 2B by the NCCN guidelines. H&P is the standard surveillance after surgical resection of pancreatic cancer based on the current NCCN and ASCO clinical practice guidelines [13,14], and serum CA19-9 test and CT are potentially beneficial to detect early recurrence after resection [19-21]. We classified the resected pancreatic cancer patients into four groups: (1) symptom group, clinical evaluation alone with H&P; (2) imaging group, combined H&P and imaging (either CT or magnetic resonance imaging); (3) marker group, combined H&P and tumor marker tests, including tests for CA19-9 and/or carcinoembryonic antigen (CEA); (4) intense group, combined H&P, tumor marker tests, and imaging.

    Because of the reimbursement policy in Taiwan, tumor markers (CEA and CA19-9) and imaging could be arranged within three months, so we defined four months as the cut-off point for classification of patients based on the follow-up into regular and irregular. We defined “regular follow-up” as clinical evaluation, including imaging or tumor markers, assessed on average every 4 months or more frequently before recurrence or within 2 years after surgery if recurrence did not occur in the first 2 years.

 Authors’ response to Q2:

We thank you for this pertinent suggestion. Accordingly, we have provided more information of the possible reasons of no survival benefit with markers or imaging monitoring and the new surveillance strategy for pancreatic cancer patients after resection in the “Discussion” section (Page 9, Lines 282-294; Page 10, Lines 295-338; Page 11, Lines 348-352) as follows:

    In the current study, we found that surveillance with combined H&P, tumor marker tests, and imaging was not associated with RFS, PROS and OS of pancreatic cancer patients who underwent curative surgery. The possible reasons are: (1) the detection methods, including serum marker tests and imaging, were not sufficiently sensitive to detect the preneoplastic lesions of recurrent pancreatic cancer early, and as reported by recent studies, circulating pancreatic cells and tumor DNA, which can be detected in the bloodstream before tumor formation, could have been helpful [25-27]; (2) the treatment methods used in the period from 2001 to 2013, including systemic chemotherapy and local chemoradiotherapy, were not sufficiently effective to eradicate the recurrent pancreatic cancer, compared to the current systemic regimens, such as FOLFIRINOX, nab-paclitaxel, and nanoliposomal irinotecan, even when the recurrent disease was detected early [28-33]; (3) the underlying tumor biology and epigenetic and genetic factors of the pancreatic cancer lessened the treatment efficacies of systemic chemotherapy, molecular target agents, and radiotherapy, and thus, contributed to tumor progression, even when the disease was detected early [34-36]; (4) the clinicopathological features and patient factors affected the prognosis of the resected pancreatic cancer [37-40], as in our study, showing the primary site as the pancreatic tail, poor tumor differentiation, and poor patient performance. Birnbaum et al. revealed that gene expression signatures are different between the head (genes more associated with lymphocyte activation and pancreatic exocrine functions) and the body/tail (genes more associated with keratinocyte differentiation) of the pancreas [40].

Pancreatic cancer is well known to be a systemic disease early in the disease course [9-11]. Therefore, detecting disease as early as possible for resected pancreatic cancer becomes important. Motoi et al. reported that persistent elevation of serum CA19-9 is a risk factor for hepatic recurrence and is associated with poor prognosis [41]. Our study also showed that patients with postoperative elevation of CA19-9 had a poorer OS. Rieser et al. demonstrated that patients with initial elevation of CA19-9 and normalization after resection, followed by persistent elevation, or those with persistently elevated CA19-9 from diagnosis through surveillance were significantly associated with poor RFS and OS compared to those without elevated CA19-9 at initial diagnosis and persistent or periodic normalization after resection [42]. Rieser et al. also showed that elevation of CA19-9 preceded the recurrence examined on CT by over 6 months. In addition to CA19-9, serum CEA can be detected early compared to detection of lesions on CT in pancreatic cancer patients after surgical resection [43]. Reitz et al. reported that a combination of CEA and CA19-9 compared to CEA or CA19-9 alone improved the prognostic prediction in OS of stage I-III pancreatic cancer patients [44]. Xu et al. showed that among pancreatic cancer patients with post-resection normalization of CA19-9, elevated postoperative CEA was an independent risk factor for poor OS [45].  Our study also showed that seven patients with postoperative elevation of both CA19-9 and CEA had the lowest OS, which was of only 7.8 months. These findings suggest that if patients have elevated CA19-9 or CEA at the initial diagnosis, using a combination of CA19-9 and CEA for surveillance may help the early use of salvage therapy if the tumor markers show persistent elevation or normalization followed by elevation, and thereby improve patient survival. Circulating tumor DNA (ctDNA) has been reported to have prognostic value [25]. Pancreatic cancer patients with ctDNA after curative surgery had shorter disease-free survival and OS [25]. The surveillance of ctDNA after resection may play a role in detecting early occult recurrence or metastases of pancreatic cancer and treating them sooner, although this requires further study to determine whether this provides a subsequent gain in survival benefit.

Panels with a combination of multiple serum markers might improve recurrence detection. Many studies focused on combination panels for pancreatic cancer screening, such as combining  carcinoembryonic antigen-related cell adhesion molecules  (CEACAMs) [46-48], osteopontin [49], or matrix metallopeptidase 7 with CA19-9 [50], or combining CEA, matrix metalloproteinases 1 (TIMP-1), and CA19-9 [51] or haptoglobin, serum amyloid A, and CA19-9 [52]. All these combination panels demonstrated improved diagnostic accuracy, but whether they could be applied to postoperative surveillance needs further investigation [53]. In addition to tumor markers, two retrospective studies showed that under CT-based surveillance for resected pancreatic cancers, patients with asymptomatic recurrence underwent more post-recurrence treatments and showed better OS compared to those with symptomatic recurrence [19,54]. These studies indicated that imaging can detect asymptomatic recurrence of resected pancreatic cancers early, and these patients have relatively good performance status and tumor biology, and thus, benefit from subsequent treatments.

We believe that the use of a combination of H&P, tests for both tumor makers (CEA and CA19-9), and CT as surveillance can detect the local recurrence or metastases in resected pancreatic cancer patients early, and thus, allow aggressive local therapy and current systemic chemotherapy regimens, which will not only increase PROS (more than our PROS of 7.8 months in the intensive group) but also prolong OS.

Round  2

Reviewer 2 Report

No further comment.